# FBG Strain Monitoring of a Road Structure Reinforced with a Geosynthetic Mattress in Cases of Subsoil Deformation in Mining Activity Areas

**DOI:** 10.3390/ma14071709

**Published:** 2021-03-30

**Authors:** Janusz Juraszek, Monika Gwóźdź-Lasoń, Dominik Logoń

**Affiliations:** 1Faculty of Materials, Civil and Environmental Engineering, University of Bielsko-Biala, ul. Willowa 2, 43-309 Bielsko-Biala, Poland; mgwozdz@ath.bielsko.pl; 2Faculty of Civil Engineering, Wrocław University of Science and Technology, ul. Wybrzeże Wyspiańskiego 27, 50-370 Wrocław, Poland; dominik.logon@pwr.edu.pl

**Keywords:** monitoring fibre Bragg grating, mining areas, strain/stress distribution

## Abstract

This paper presents implementation of purpose-designed optical fibre Bragg grating (FBG) sensors intended for the monitoring of real values of strain in reinforced road structures in areas of mining activity. Two field test stations are described. The first enables analysis of the geogrid on concrete and ground subgrades. The second models the situation of subsoil deformation due to mining activity at different external loads. The paper presents a system of optical fibre sensors of strain and temperature dedicated for the investigated mattress. Laboratory tests were performed to determine the strain characteristic of the FBG sensor-geogrid system with respect to standard load. As a result, it was possible to establish the dependence of the geogrid strain on the forces occurring in it. This may be the basis for the analysis of the mining activity effect on right-of-way structures during precise strain measurements of a geogrid using FBG sensors embedded in it. The analysis of the results of measurements in the aspect of forecasted and actual static and dynamic effects of mining on the stability of a reinforced road structure is of key importance for detailed management of the road investment and for appropriate repair and modernization management of the road structure.

## 1. Introduction

The analysis of the use of systems for long-term monitoring of the state of a road structure attracts a lot of attention, especially if the construction investment is located in an area of III geotechnical category with impacts on mining activity. By providing precise real-time information on the structure and its surroundings, it is possible to evaluate the state of the structure using diagnostic and prognostic tools based on the available data. The separation of intrinsic features from comprehensive monitoring data would be the key task in the assessment of the structure state and, consequently, an effective verification of the assumed unfavourable effects of mining activity compared to real impacts. The reasons for the use of components with optical fibres as sensor elements in the monitoring of the state of structures in use are well known.

The ever-growing demand for new linear investments usually involves improvement to land with difficult geotechnical conditions. Such land is often described as low-bearing soils, including the herein analysed soils in areas affected by mining impacts [1,2]. The basic mechanisms determining the engineering methods of computational analysis of subsidence are connected with phenomena such as compaction, consolidation, changes in water conditions, or disturbances to the balance of soil massif. The road structure monitoring investigated in the paper is essential during the use of a construction investment to reduce factors generating investment risk.

Structures located in mining areas are subject to additional impacts due to surface deformation or subgrade/subsoil vibration. These impacts are transferred from the subsoil to the structure, which results in the occurrence of strains and internal forces in the structure. The monitoring of the strains provides current insight into the structure operation and creates the possibility of current checks on individual limit states of the structure.

In the case of an earthen structure, the strain-related impact of mining affects a significant portion of its area. In the zone of tensile strains, the subsoil and the embankment are loosened, and large limit-state areas are created. This leads to uneven settlement of the embankment crown, local subsidence, and horizontal displacement of the road pavement, as well as to local slides of slopes and pavement damage posing a hazard to safety. The presented monitoring system for a selected road structure scheme offers completely new possibilities for the use and protection, as well as renovation and modernization of road structures. In areas with mining impacts, the road structure subgrade is reinforced to increase the load-bearing capacity, reduce the subsidence of structures, prevent the loss of stability in the form of slips or landslides, and prevent the subgrade from liquefaction. Stabilization of the subgrade structure and mitigation of the effects of mining-related deformation were observed during numerous tests.

While choosing the method of subgrade reinforcement, the land development features should also be taken into account because the application of some methods, e.g., dynamic consolidation, may have a negative impact on existing building structures (resulting in damage to buildings due to transmission of vibrations from the subsoil or changes in hydrogeological conditions)—this is another thing that confirms the necessity and usefulness of monitoring in this field.

The data collected from the conducted analyses of selected models of the reinforcement of subgrades affected by the impact of mining activity additionally create the approach to the use of anthropogenic ground as a material for the proposed geosynthetic mattresses. This type of ground has already been tested in terms of its physical and strength parameters, as well as in the environmental aspect–utilization of materials from postmining waste dumps [3]. A database is being created for the environmental analysis of the issue.

Monitoring is an interdisciplinary issue. Due to its usefulness, it has a wide range of applications in the analysis of direct and indirect variables affecting variables related to the indicated aspects: securing the interests of the investor and the contractor of a construction investment, investment risk, especially in areas with the impact of mining activity, optimal assurance of the safety of investment implementation and use, and current assessment of the technical condition of a linear structure during its operation, as well as control and the possibility of getting an early warning of deformation of engineering structures in areas with an adverse mining impact. The presented empirical approaches to the monitoring of the interaction between the subsoil and the road structure provide a great deal of data that can be properly managed in a modern approach to the structure repair and modernization management. The monitoring data can generate appropriate models of the support of the decision-making process with methods suitable for the above-specified description of differentiating factors from the perspective of the researcher and analyst. The monitoring type and methodology and the range of the obtained data continuity for the time variable generate whole sections of statistics which can be used to find the answer to a given design-, strength-, or usability-related question, as well as economic, operating, or environmental problems. In this publication, the obtained data are analysed in the aspect of the road structure protection against the mining impact using a geomattress [4,5]. For the set of data divided into main structural, protection, and reinforcement elements and monitoring measurement locations, a data-mining analysis was conducted to explore large data resources in search of systematic correlations between variables and then to evaluate the results by applying the detected patterns to new subsets of data.

The final goal of the data mining operations on the data coming from the created monitoring system is to predict the behaviour of the road structure and provide timely feedback to prevent the possibility of increasing the probability of the road structure damage. Predictive data mining gives direct economic benefits in the management of road use and renovation. The data from individual monitoring models can be used to build a model for specific patterns, together with evaluation and verification to obtain predicted values or classifications.

Many various monitoring technologies have been developed, including the sensor based on the optical fibre Bragg grating (FBG). This solution is now being investigated in terms of its possible application for road structures, herein for example in the area affected by mining activity. In 1966, the use of optical fibres for digital transmission of data was proposed by Kao, who later won the Nobel Prize for Physics in 2009.

This paper presents the results of testing the use of FBG sensors for the monitoring of the reinforced surface of a road structure affected by unfavourable static and dynamic impacts due to mining activity. The aim of the investigated monitoring system is to develop guidelines for the analysed variable that affect the comprehensive manner of utilization, operation, and management of a road structure in areas affected by tremors and subsidence due to mining.

This type of monitoring was recently investigated in [6,7,8,9,10,11] and [12]; proposals for the use of the results of already tested monitoring systems to analyse the mining activity impact on residential and commercial buildings and road structures are signalled in [13,14] or [15]. The strain measurement method based on the change in the wavelength on the Bragg grating is described in [2,8,16,17,18,19].

As it is known, the Bragg grating is defined as periodic disturbance of the effective absorption coefficient and/or the optical fibre refractive index. Based on that, quasi-distributed systems are created for the detection of structural hazards. Herein, it concerns a road structure. The operation of mines, both past and present as well as future, generates imbalance of the rock mass. On the surface of the area, this results in direct effects, both continuous and discontinuous, indirect effects, and dynamic effects. All these effects have an unfavourable impact on buildings and road structures.

The main aim of the work is to use innovative FBG sensors to determine real values of strains and forces occurring in the geomattress depending on different service loads.

## 2. Presentation of the Problem for the Monitoring System

The aim of the testing and analyses presented herein is to demonstrate an original approach to the monitoring of changes in the stress-and-strain state of the mining subsoil for the reinforcement subsoil layer directly interacting with the construction of the road in areas with mining impact. The development of the civil engineering sector is directly linked to the development of the technology generating new solutions for the phases of design, realization, and operation of a construction investment.

The scope of innovations for the monitoring system creates new data and new schemes of analysis, e.g., for the investment risk calculation in the management of a road structure in an area affected by mining damage. Due to a continuous measurement of specific physical quantities, it is possible to control the behaviour of a civil engineering structure [4,5,15,20,21]. The authors point to the capabilities of a system that:
(i)identifies the actual behaviour of the subsoil under designed constant and variable loads as well as under the forecasted and unpredictable impact of mining activity;(ii)provides data for online description of the subsoil–reinforcement element–road structure interaction for the analysed period of utilization and operation of a civil engineering structure;(iii)is an important variable in the optimization of schedules of the structure repair and modernization and in the checking of the investment risk value depending on the variable related to the mining activity effect.

The guidelines for road design and construction binding in the EU require that earthen structures as well as surfaces should be designed and constructed so that any potential impacts and influences occurring during construction and use, including the effects of mining activity, should be carried appropriately. In these specific conditions, the structures should display adequate durability, taking account of the predicted service life, and should not succumb to destruction to an extent disproportionate to the cause. Meeting these requirements is equivalent to the need to ensure conditions in which load or usability limits are not exceeded not only in each individual element but also in the entire earthen structure together with the road surface.

The nature of damage to the road infrastructure in mining areas and of the mobilization of limit states of load capacity is of a completely different origin compared to other areas [1,4,22,23,24] and is the effect mainly of the susceptibility of earthen structures and surfaces to horizontal unit strain with a loosening character (ε (mm/m)). This problem is illustrated comprehensively in the diagram in Table 1 and Figure 1, Figure 2 and Figure 3.

At present, the majority of right-of-way structures in the road infrastructure still do not demonstrate sufficient structural resistance to such destructive impacts, which is the main cause of their damage. As stipulated by the regulations now in force, areas affected by mining activity should be protected according to the category of the mining area. The current classification of mining areas is presented, taking account of continuous and discontinuous deformation, paraseismic impacts and postmining areas. The measure of the hazard posed in a mining area by dynamic impacts both to newly erected and already existing structures is the assignment of a specific seismic zone to the area [25,26] and [27,28]. The seismic zone is described by parameters of the maximum ground vibrations that can occur in the area: acceleration and speed, as well as the subsoil design acceleration, all of which have to be taken into account in the design of civil structures in the area.

The indirect impacts affecting structures and related to the drainage of tertiary layers in the form of a large-size drainage basin are described using a single parameter: the mining area subsidence due to drainage. In areas where the impacts are not related to ground subsidence caused by mining, they do not result in mining damage to the development of the land or mining damage of a hydrogeological nature that is attributed in particular to changes in the stress-and-strain state of the ground. The definition of the mining area is directly connected to the range of the road structure interaction with the subsoil subjected to strain. According to ref. [29], this is the mass rock layer close to the surface, which is usually built of different soils, where the impact of the structure on the stresses and strains arising in the layer are considered as essential. The analysis of the results presented herein is based on the method of the Budryk-Knothe geometric-integral theory used in the majority of geomechanics in Silesia for prognostic analyses. It is the basic method of identification of kinematic limit states of investigated structures.

The analysed issues concern mining areas with subsoils reinforced with a geomattress to protect the right-of-way structure against deformation due to mining activity. The measure of the hazard posed to a mining area by impacts causing deformation is the mining area category. The category is described by indications of deformation. It is a variable that substantially affects the risk factor of the safe use of the structure. The basic indicators are subsidence—w, horizontal strain—ε (mm/m), land slope—T (mm/m), and curvature radius—R (km).

## 3. Experimental Setup

### 3.1. Test Stand Modelling

Two test stands were designed and made to achieve the research goals. The stands are equipped with optical FBG sensors of strain and displacement enabling a continuous measurement of strain in selected points of the geogrid. Test stand 1 is composed of the following layers: a 15 cm sand bed and the load-bearing mattress made of a geosynthetic grid filled with 16–32 mm aggregate with the thickness of 50 cm, compacted in layers every 25 cm. It is based partially on subsoil and partially on a reinforced concrete slab, which allows a comparison of strains in the two parts. Each subgrade has a separate FBG strain sensor embedded in the geogrid. The diagram of the test stand is shown in Figure 4.

Test stand 2 enables an analysis of the case when there is no subgrade under the geogrid in two strips. It is composed of the following layers: geotextile, 800 × 1200 wooden pallets, and a load-bearing mattress filled with 31.5–63 mm aggregate with FBG sensors and 31.5–63 mm aggregate. The diagram of the test stand is shown in Figure 5.

Two types of FBG sensors are applied: the strain sensor and the displacement sensor. Due to high values of the mattress deflection angles, the strain sensor proved useless and was no longer used in further stages of the research. On a single measuring line, the strain sensor may have 10 sensors with the wavelength difference of 5 nm with temperature compensation. A special technology of gluing the sensor into the geogrid in the middle of its span was developed. The sensor and its parameters are presented in Figure 6.

The strain of the optical fibre sensor is the function of the wavelength (1) measured by the optical FBG-800 interrogator enabling dynamic measurements. The sampling frequency is 2000 Hz,
(1)$$\Delta \varepsilon =\frac{\frac{\lambda_{\mathrm{act},\mathrm{strain}\;}-\lambda_{0,\;\mathrm{inst},\mathrm{strain}\;}}{\lambda_{0,\;\mathrm{inst},\mathrm{strain}\;}}-B\times \left(T_{\mathrm{act}\;}-T_{0,\;\mathrm{inst}}\right)}{A}$$
where A and B are constant Table 2.

### 3.2. Plan of Investigated Experiments

The testing was divided into two main parts: the laboratory stage and in situ testing. The laboratory stage included the process of calibration of the geogrid strains with the forces occurring in the grid. The in situ testing was carried out both during the construction of the test stand and during the load tests. The test stand preparation was divided into individual stages, which had a considerable impact on the geogrid strains Table 3.

The results of the geogrid strain-state testing during the construction of test stand 2 are presented in Figure 7. The first stage of laying out the geogrid produced strains at the level of about 135 μstrain. The jump in the graph at 14:04:47 h to the value of 600 μstrain proves that the geogrid backfilling process began. Backfilling was performed by hand using shovels. It lasted about 15 min, which can be seen in the graph to 14:19:00 h. The next stage was the geogrid forming and wrapping up to shape the geomattress. During that time, the geogrid strains were constant and totalled about 750 μstrain. The final stage of the test stand preparation was filling the geogrid mattress with aggregate, which can be seen in the graph between 14:32 and 14:43 h. The works were finished at 14:44 h, which can be observed in the graph in the form of a stop to the increase in strains. Strain values then stay at the level of 1050 μstrain. This is the reference level for further load tests. In the initial stage an analysis was also conducted of strains arising during the aggregate compaction with a compactor. The aggregate was compacted separately over each type of subgrade (subsoil, concrete slab).

Distinct jumps in strains can be observed in Figure 8 when the compactor was directly over the sensor. After the compactor moved beyond the sensor, the strain value decreased at once. The graph also indicates that the geogrid mean strain rose cyclically. Characteristic steps (red line) appeared due to subsequent stages of the stand compaction, from 12 to 21 μstrain. Compaction of the aggregate over the part located on the natural soil resulted in different values of the geogrid strains. Distinct jumps in strains can be seen in Figure 9 when the compactor was directly over the sensor.

After the compactor moved to other places, the strain value in the tested location decreased considerably. After each compaction cycle, the geogrid suffered no plastic strain, which is illustrated by the red line. The natural soil thus plays the role of an elastic subgrade.

## 4. Results

The following results were obtained for individual stages of the testing for the designed monitoring technology. The technology was suited to the initial data of the investigated issue concerning the analysis of un-forecasted mining activity impacts on right-of-way road structures—roads with appropriate reinforcement of the subgrade using a geosynthetic mattress.

### 4.1. In Situ Testing Results

In situ tests were performed including: (a) static loading of the area under the FBG monitoring, (b) dynamic loading of the tested structure, (c) loading due to the use of construction equipment–88kN ascent of an excavator, and (d) putting the excavator supports on the area of the FBG monitoring. The second experiment performed on the test stand was a static test consisting in loading the structure with concrete slabs. Standard 50 × 50 × 7 cm paving slabs, each producing a 350 N load, were used for this purpose. The load was applied in layers with four slabs each. The weight of each of the layers produced a 1.4 kN load. The load was applied in places where the FBG sensor was installed. Moreover, in places under which there was a layer of pallets, an additional load was applied to additionally ballast the pallets to make the forces from the proper load to be taken over by the geogrid mainly. This additional load was not included in the weight of the layers producing the proper load. The experiment was carried out in stages by laying subsequent layers of slabs onto the stand. In the final stage of the testing, there were five layers of slabs on the stand. The total weight of the load was then 7.0 kN. The testing results are presented in Figure 10.

For one layer of slabs—the weight equal to 1.4 kN—the strain values totalled about 70 μstrain. After the second layer was laid, when the weight rose to 2.8 kN, the strain level in the geogrid increased to 130 μstrain. The next layer of slabs caused a further rise in strain to the value of 240 μstrain. After the fourth layer of slabs was laid, the load totalled 5.6 kN, and the strain level rose to 380 μstrain. Lastly, the fifth layer of slabs resulted in another rise in strain to the value of 450 μstrain. The next experiment carried out on the field test stand was the free-fall test of concrete slabs simulating a dynamic load of the type of the Heaviside step function. Next, 50 × 50 × 7 cm concrete slabs were dropped from different heights of 30, 60, 90, and 120 cm, respectively. The idea of the testing is presented in Figure 11, and the testing results are shown in Figure 12.

The free fall of the slabs from different heights caused jump changes in the strain value. At a free fall from the height of 30 and 60 cm, the slab caused a strain jump to the value of 600 and 1000 μstrain, respectively. The slab free fall from 90 and 120 cm caused a jump in the geogrid strain value to 1400 and 2050 μstrain, respectively. This means that the impact of falling slabs caused a jump change in the values of the geogrid strains. The geogrid then returned to the initial state. In the graph it can also be noticed that after the slab free fall from 90 cm the geogrid plastic strain occurred and the strain level being the result of the next impact totalled 100 μstrain. In addition, in this experiment with impulse loads, the FBG sensors performed very well. They made it possible to analyse the geomattress strain state and provided an answer to the question whether or not the mattress had been damaged and whether or not the loads had produced plastic strain values. The next experiment performed on the field test stand for the geogrid monitoring was the run of the CASE Super R backhoe loader. The machine mass totals 8850 kg. The backhoe loader ran onto and left the stand three times. The loader run was controlled to ensure that its rear wheel was directly over the sensor. The loader ran onto the stand to cause maximum strain of the geogrid. Figure 13 shows the geogrid strain testing results during the loader run onto the test stand.

During the first run of the loader the maximum strain value totalled 1200 μstrain. It then dropped to the level of −200 μstrain to reach during the next run the maximum value of about 1380 μstrain. During the break between the loader second and last run onto the test stand, strains fell to the value of −300 μstrain. The last run produced a jump in strain to the value of 4300 μstrain. After the loader left the stand, the strain values dropped again to the level of −370 μstrain. This is the response of the geomattress system to its considerable strain.

The last experiment on the test stand was also carried out using the CASE Super R backhoe loader. This time the excavator supports were spread out and raised the machine. The supports were placed over the sensor to produce the highest loads of the area covered by the FBG monitoring and obtain the highest strains of the geogrid. The strains arising when the construction machine was placed on the geogrid are presented in Figure 14.

Raising the loader on the supports caused an increase in the geogrid strain. After the first raising the strain level totalled 1780 μstrain. When the loader left the stand, the strain values decreased almost to zero to reach the level of 1650 μstrain at the next raising. After the loader left the stand, the strain level dropped again to the value of 50 μstrain. In the last test of raising the loader, the obtained strain value totalled 1770 μstrain. The drop in strain below zero from 10:43:42 h proves that the loader left the stand.

### 4.2. Laboratory Testing Results

The aim of the laboratory testing was experimental determination of relations between strains and loads (axial forces) in a unidirectional grid. For this purpose, a series of tensile tests of specimens from the geosynthetic grid were performed. The reference to the standard force was the dynamometer of a strength-testing machine (HOUNSFIELSD TEST EQUIPMENT 0133 model H50KS, Shakopee, MN, USA with Accuracy Class 0.5 (Figure 15). The geogrid was the same as the one used in the field stand structure. The rupture test was performed according to standard EN ISO 10319:2008 geosynthetics–wide-width tensile test. Thirty (30) specimens 200 mm wide and 200 mm long were prepared for the testing to ensure the nominal length of 100 mm according to 6.3.3 PN-EN ISO 10319:2008. As the geogrid is reinforced unidirectionally, the specimen was ruptured in the longitudinal direction only. Special holders—elastomeric inserts—were designed to hold the strength-testing machine, which ensured that the loading process was carried out correctly. Subsequent reference testing was carried out in the same manner, except that the geogrid was equipped with an optical FBG sensor permanently glued into the grid using special technology.

Such embedding of the sensor into the geogrid worked very well in the case of field testing. The FBG sensor was glued into a 620-mm-long geogrid specimen to ensure the nominal length of 500 mm. Rubber inserts were fixed at the ends of the specimen to ensure the best assembly. The inserts were joined to the geogrid using a cyanoacrylate-based adhesive. The results of calibration tests are presented in Figure 16.

The obtained calibration chart makes it possible to determine the value of the load occurring in the geogrid based on the measurement of the grid strain using an FBG sensor.

## 5. Discussion

The paper presents an innovative method of measuring strains of the geosynthetic grid using optical FBG sensors. The strain measurement, together with determination of the forces acting in the geogrid, is a very important element of the monitoring of the condition of a geotechnical structure and areas affected by mining damage. The values of strains and loads occurring in the geogrid were determined experimentally. The stand construction and the backfilling with aggregate caused strains at the level of 1050 μstrain corresponding to the load of 220 N/m. This strain state was adopted as the initial one. The highest strain values were caused by dynamic loads in the form of the Heaviside impulse of 2050 μstrain comparable to the excavator ascent onto the mattress. The lowest strain values of 70 μstrain were recorded during aggregate compaction on natural soil. The optical FBG sensors belong to the family of the fibre-optic sensors (FOS), and they perform very well in difficult terrain and environmental conditions. Individual FBG sensors can be connected easily to obtain a comprehensive measurement system of the entire structure. The sensors can be used to measure the structure strains on the one hand and the temperature of the structure or the moisture level on the other. However, their most important task is the strain measurement as strains are the factor that has the most essential impact on the safety of the use of a structure.

A very important value of the work is determination of real values of the geogrid strains and real values of forces acting in the grid. This is important as it creates the possibility of the geogrid stress-and-strain state evaluation. The works published so far do not refer to the values of forces in the geogrid and do not enable the grid stress-and-strain state evaluation. Most of them concentrate on monitoring the condition of a few key geotechnical structures, including soil nail systems, slopes, and piles, without any reference to the issue of the stress-and-strain state in the analysed elements. Ref. [30] points to the possibility of using the results of the grid strain measurement as input or boundary data for a numerical experiment using the finite difference or the finite element method. This will be the subject of the authors’ further work. Many works concentrate on the FBG sensor mounting method. Two solutions can be distinguished. The first consists in applying an optical fibre with Bragg gratings over an intermediate and properly secured element. This works in structures with low deformability and is not proposed here because it would generate too large a measurement error. The second solution is to embed the optical fibre sensor in the tested structure using an adhesive. A special gluing technique was used in the analysed geogrid to embed in it not only the FBG strain sensor but, significantly enough, also a telecommunication optical fibre to ensure data transmission and the sensor supply with light. Indicating directions of further development, a proposal is made in [29] to construct 3D sensors. The research results point to the possibility of constructing orthogonal strain-measuring systems, 90- and 120-degree strain rosettes, as well as 3D systems. Another issue raised in to-date publications is the cost of the FBG monitoring system [30]. The experimentally determined strain values [19,31,32] are comparable to the values occurring in the geogrid. Undoubtedly, monitoring systems based on optical FBG sensors are relatively expensive. Low-cost solutions are being sought. The authors tested a new generation of 1 Hz optical interrogators with FBG sensors operating at the wavelength of 800 nm. Such a solution is equivalent to sensors working at the wavelength of 1550 nm and, most importantly, is twice as cheap as classical solutions. Attention should also be drawn to the unprecedented possibility of integrating different fibre-optic sensors to perform parallel measurements of strains, the forces occurring in the geogrid and the current temperature and moisture content. Another advantage of measurements based on optical fibre sensors is their fatigue strength, long service life, and the multiplexing capacity.

## 6. Conclusions

Appropriately processed, the data obtained from the geogrid monitoring can become the database for the analysis of risk presented by the operation of a structure in cases of un-forecasted mining activity impacts. Known forces of the interaction along the mining-activity–reinforcement-system–road-structure scheme in the monitored time interval should considerably improve the management of road structures in areas affected by unfavourable impacts of mining activity. The probability defining the risk factor related to the use of a road structure is usually expressed as frequency, i.e., the number of expected events in a unit of time. For structures provided with appropriate monitoring and covered by permanent analysis of variable values qualifying the structure for specific mining area categories, the frequencies of unfavourable and random events may vary by many orders of magnitude for the analysed logarithmic scale. The categories of the consequences of on-line monitoring of the work of a road structure can also be expressed in many matrices of risk related to the safety of the structure. The assumed schemes of data analysis give theses with very good results. Subsequent planned testing of buildings and structures in areas with forecasted mining impacts can become the basis for the verification of both the adopted monitoring methodology and the system of analysis of obtained data in the aspect of risk related to the use of an investment and optimization of schedules with the scope, type, and frequency of the performed repair and modernization management of the structure. Moreover, data are collected all the time for cost analysis of such monitoring of the structure. For the preliminarily developed risk matrices for selected types of mining damage repair work taking account of probability and consequence classes, the initial results are very promising. The combination of optical FBG sensors with geosynthetic materials and their application in road structures make them undoubtedly one of the innovative methods of strain measurement, and the presented calibration methodology enables the assessment of loads occurring in the geogrid.

Summarizing the initial assumptions, the adopted models, and the obtained final results, together with a discussion of current research trends, the following main points can be mentioned:
The applied system of FBG sensors enables precise and continuous monitoring of strains and loads of the road structure reinforcement in the form of a geomattress which takes over the impacts from the mining activity areas.Compared to the existing systems, the presented monitoring system is characterized by exceptional resistance to difficult environmental conditions.In the future, the monitoring information package can become the basis for a system warning against mining hazards presented to existing engineering structures. Such information is now indispensable for statistical analyses of the ground behaviour in the interaction with an engineering structure.

The growing set of monitoring data should be analysed in all interdisciplinary issues that are connected with the discussed monitoring system intended for road structures in areas affected by the impact of mining activity. Datasets are being created for a detailed environmental, economic, and performance analysis of the investigated investment in the form of a monitoring system intended for a road structure reinforced with a geosynthetic material.

## Figures and Tables

**Figure 1 materials-14-01709-f001:**
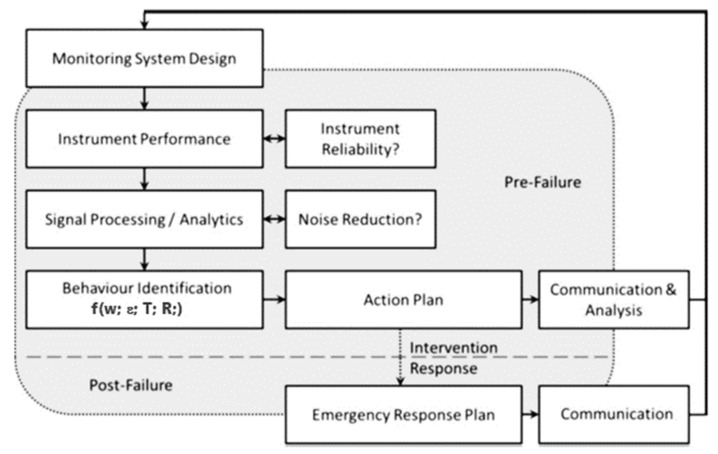
Scheme for the road structure monitoring process for the implementation of the safe operation process with the assumed investment risk.

**Figure 2 materials-14-01709-f002:**
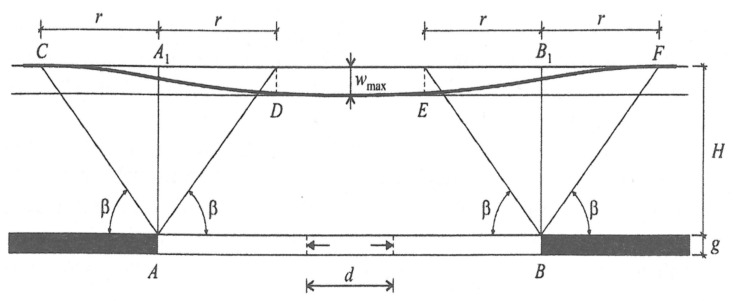
Deformation of the surface of the analysed land properties by factors affecting the characteristic of the range of mining impacts. Fixed mining trough in a plane deformation state: β—value of the main influence range angle; w_max_—maximum lowering of the mining trough (m); g—thickness of the selected layer (m); a—service factor for w_max_ = a ∙ g

**Figure 3 materials-14-01709-f003:**
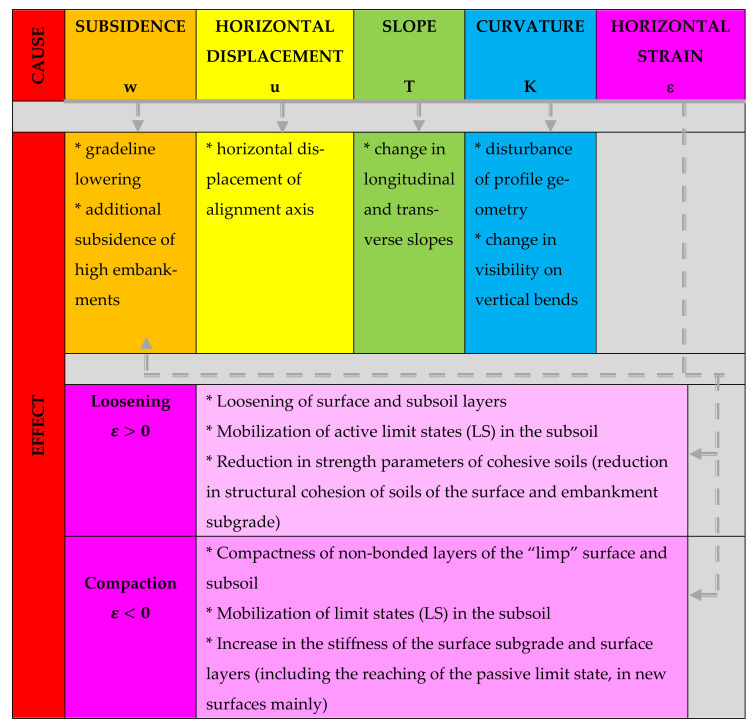
Cause-and-effect relationship describing the impact of indicators of the mining area continuous deformation on functional and strength parameters of the road infrastructure.

**Figure 4 materials-14-01709-f004:**
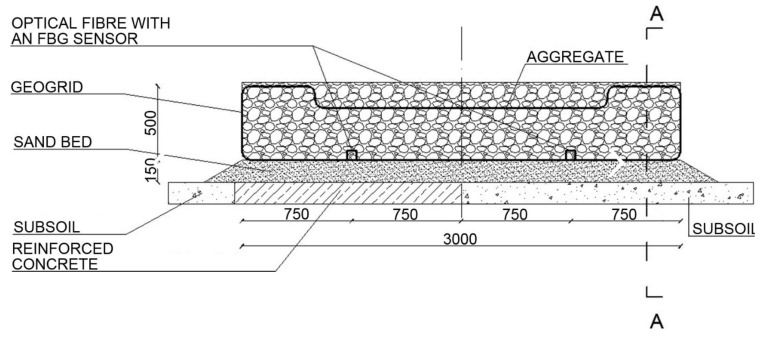
Diagram of test stand 1 (cm).

**Figure 5 materials-14-01709-f005:**
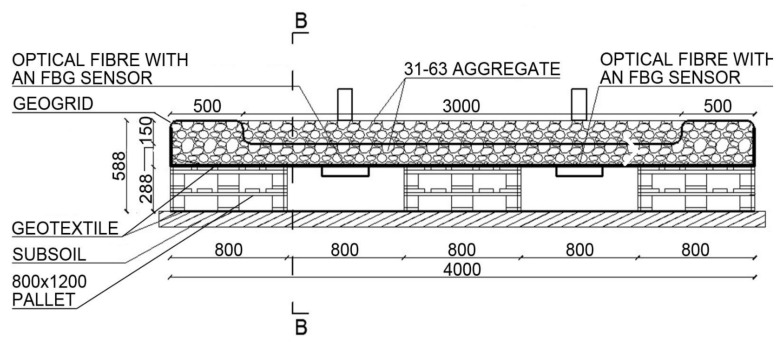
Diagram of test stand 2 (cm)**.**

**Figure 6 materials-14-01709-f006:**
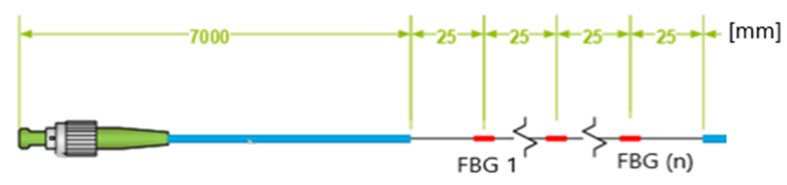
Scheme of fibre Bragg grating (FBG) sensor (mm)**.**

**Figure 7 materials-14-01709-f007:**
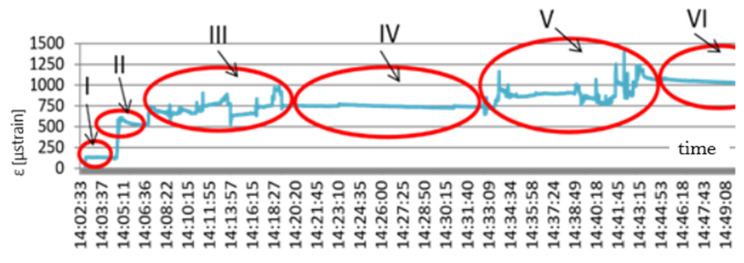
Scheme geogrid strains—stage zero of the monitoring: x (s); y (μstrain).

**Figure 8 materials-14-01709-f008:**
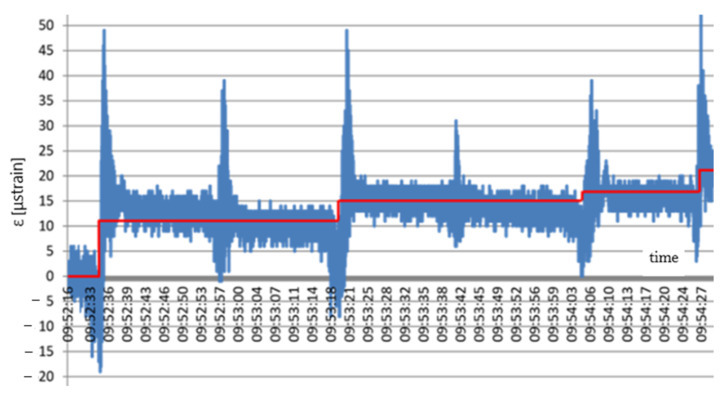
Scheme strains in the geogrid based on a concrete slab: x (s); y (μstrain).

**Figure 9 materials-14-01709-f009:**
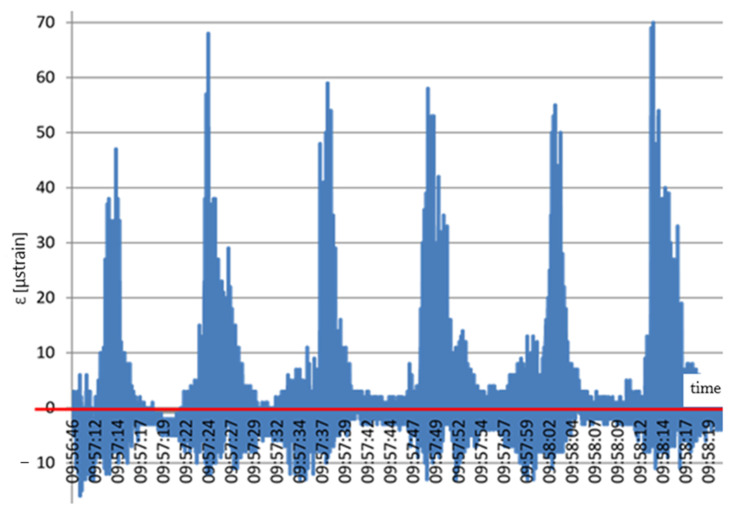
Scheme geogrid strains arising due to the interaction between the subsoil and the aggregate compaction process: x (s); y (μstrain).

**Figure 10 materials-14-01709-f010:**
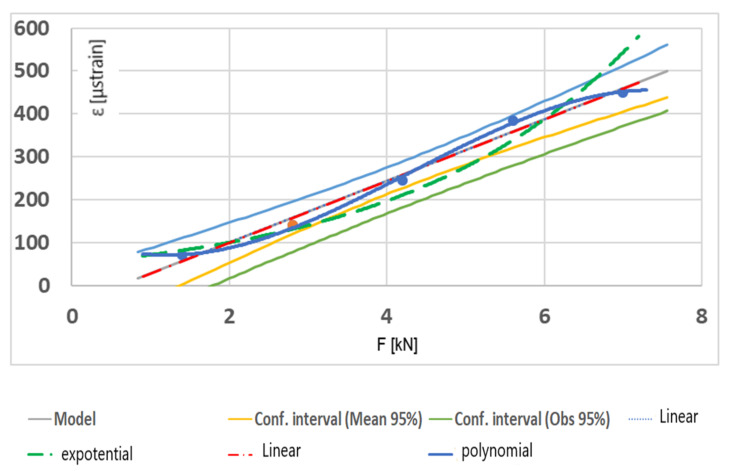
Geogrid strain depending on static load.

**Figure 11 materials-14-01709-f011:**
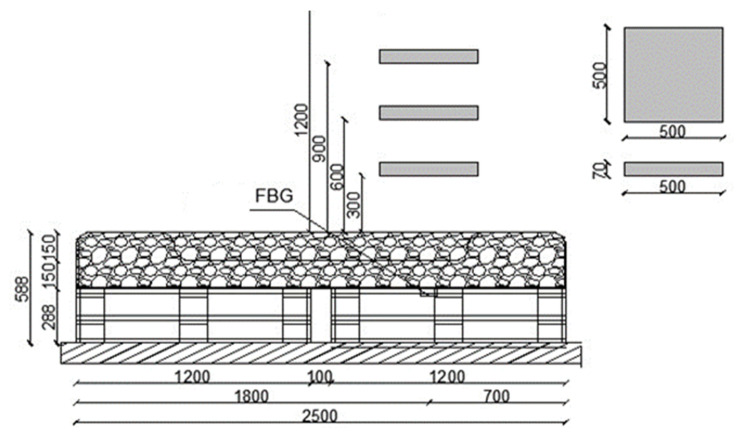
Idea of the testing—Heaviside step function (cm).

**Figure 12 materials-14-01709-f012:**
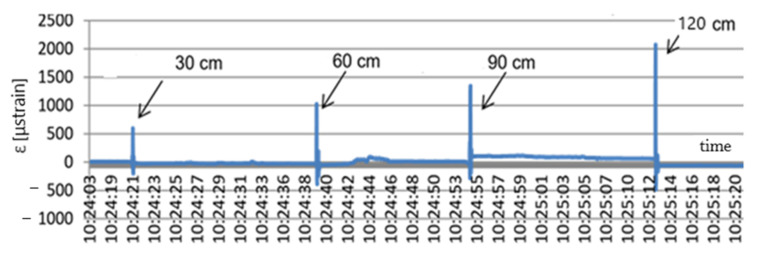
Dynamic test results: x (s); y (μstrain).

**Figure 13 materials-14-01709-f013:**
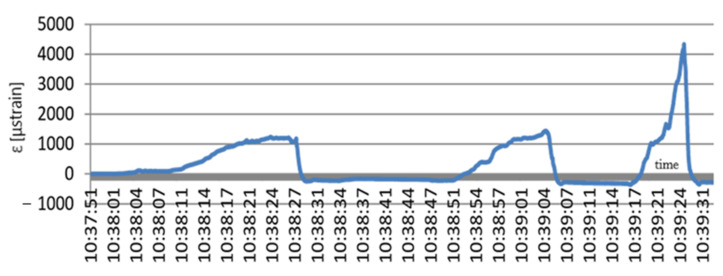
Geogrid strains during the backhoe loader run; x (s); y (μstrain).

**Figure 14 materials-14-01709-f014:**
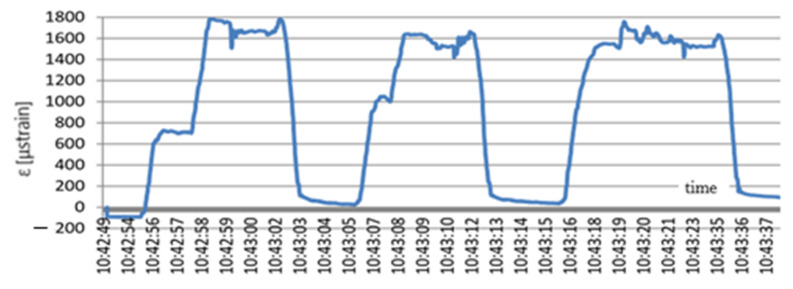
Change in the geogrid strain values due to the impact of the loader weight; x (s); y (μstrain).

**Figure 15 materials-14-01709-f015:**
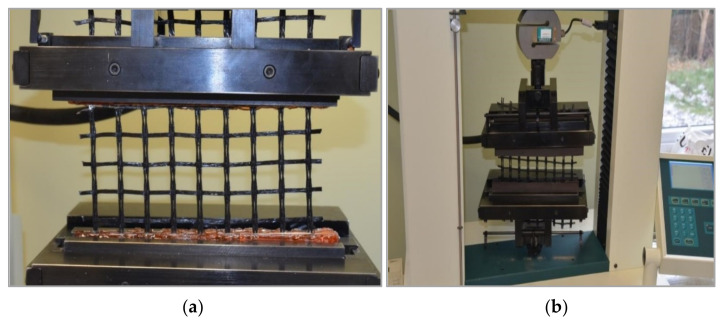
Geogrid tensile test: (**a**) fastening of the geogrid, (**b**) force sensor.

**Figure 16 materials-14-01709-f016:**
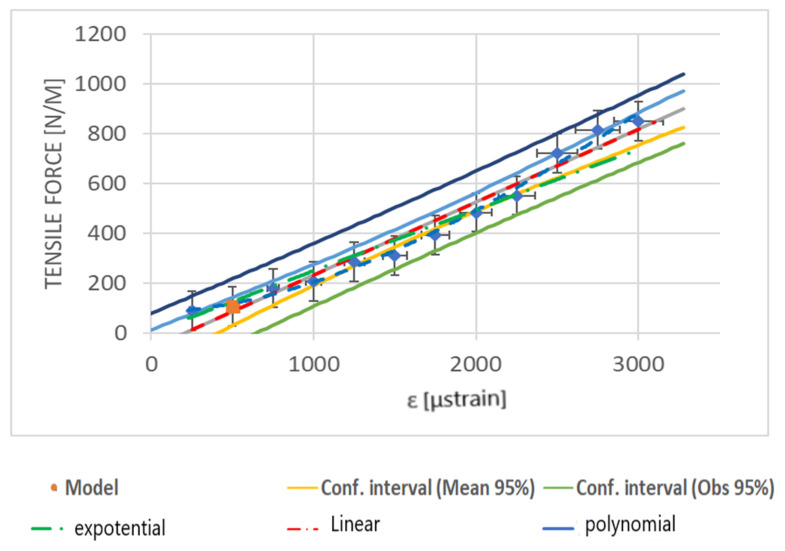
Calibration testing of the geogrid with an optical FBG sensor.

**Table 1 materials-14-01709-t001:** Classification of mining areas in the area of continuous deformation indicators of land deformation.

Mining Area Category	Slope T (mm/m)	Curvature Radius R (km)	Horizontal Strain ε (mm/m)
0	T ≤ 0.5	|R| ≥ 40	|ε| ≤ 0.3
I	0.5 < T ≤ 2.5	40 > |R| ≥ 20	0.3 < |ε| ≤ 1.5
II	2.5 < T ≤ 5.0	20 > |R| ≥ 12	1.5 < |ε| ≤ 3.0
III	5.0 < T ≤ 10	12 > |R| ≥ 6	3.0 < |ε| ≤ 6.0
IV	10 < T ≤ 15	6 > |R| ≥ 4	6.0 < |ε|≤ 9.0
V	15 < T	4 > |R|	9.0 < |ε|

**Table 2 materials-14-01709-t002:** The strain of the optical fibre sensor is the function by the optical FBG-800 interrogator enabling dynamic measurements A and B.

Measurand	Description	A (με^−1^)	B (°C^−1^)
Δε (με)	Strain shift	7.7584 × 10^−07^	5.8929 × 10^−06^
λ_0,inst,strain_ (nm)	Initial strain wavelength		
T_0.inst_ (°C)	Initial temperature		
T_act_ (°C)	Actual temperature		
λ_act,strain_ (nm)	Actual strain wavelength		

**Table 3 materials-14-01709-t003:** Stages of the preparation of the test stands for the monitoring of the assumed road structure with a reinforcement element in the form of a geosynthetic mattress.

Stage	Scope of Works Leading to the Construction of the Test Stand
I	laying out the geogrid—no load
II	commencement of backfilling
III	mattress formation—first aggregate layer
IV	mattress formation—wrapping up the geogrid
V	mattress formation—second aggregate layer
VI	completion of the test stand construction
VII	tampering

## Data Availability

No new data were created or analyzed in this study. Data sharing is not applicable to this article.

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
