# Peer review of "FBG Strain Monitoring of a Road Structure Reinforced with a Geosynthetic Mattress in Cases of Subsoil Deformation in Mining Activity Areas"

_materials, 2021, doi:10.3390/ma14071709_

Round 1

Reviewer 1 Report

The paper proposes a monitoring system using optical FBG sensors embedded into the structure to determine values of strain in reinforced road structures in areas of mining activity. The employment of optical FBG sensors for this purpose seems very natural, and the presented results show that the system is able to distinguish different impacts. I recommend the paper for publication. Before publication, figures should be carefully revised to represent legend accurately, e.g. "XSTAT trial" should be removed from Fig. 10, resolution of some figures (3-6, 8-9) should be also improved, legends in Figs. 10, 16 are suggested to be represented in a full form.

Author Response

Those comments are all valuable  and  very  helpful  for  revising  and  improving  our  paper,  as  well  as the   important   guiding   significance   to   our   researches. We have studied comments carefully and have made correction which we hope meet with approval. Revised portion are marked in red, green or blue  in the paper.

Kindly informs that, as suggested by reviewers, the article has changed and added the following:

- the introduction point has been developed - reviewer no.1

- the discussion point and critical comments were developed - reviewer no.1 and no.2

- a new Experimental Setup point has been created with the appropriate sub-points as suggested by reviewer no.3

- citations have been corrected

- wrote to the DOI literature

- the drawings have been corrected in line with the reviewer's suggestion no.3

- as suggested in review no. 3, the indicated text was listed in accordance with the editorial form

- Figures 4, 5 and 6 have been enlarged as indicated in review 3, as far as the editorial form of the publication allows

Reviewer 2 Report

The paper is interesting and important. Some comments can be found below:

-We live now in a climate emergency so its most strange that the authors have not start the paper by mentioning exactly that. It seems that they are not aware about the words of a Professor of Physics at the University of Oxford authored a paper where one can read the following:

 “Let’s get this on the table right away, without mincing words. With regard to the climate crisis, yes, it’s time to panic”

Pierrehumbert, R., 2019. There is no Plan B for dealing with the climate crisis. Bulletin of the Atomic Scientists, pp.1-7.

So please start the introduction by draw a connection between environmental degradation, and resource efficiency.

-Comments on the importance of infrastructure durability are also needed

- The literature review needs to be improved. Too many references are authored by Polish academics meaning that the most relevant literature in the field as not been cited. That way is not possible to assess if the current investigation will help to fill a research gap in international literature

Author Response

(The authors gave the same response as above.)

Reviewer 3 Report

Please find attached a PDF file with my comments and suggestions for authors.

Author Response

(The authors gave the same response as above.)

Round 2

Reviewer 2 Report

The authors have improved the paper

Reviewer 3 Report

My comments and suggestions have been addressed. Therefore, in my opinion the manuscript can be accepted for publication.